

# The Boundary Layer Dispersion and Footprint Model: A fast numerical solver of the Eulerian steady-state advection-diffusion equation

Mark Schlutow[1], Ray Chew[2], and Mathias Göckede[1]

[1]Max Planck Institute for Biogeochemistry, Jena, Germany
[2]California Institute of Technology, Division of Geological and Planetary Sciences, Pasadena, CA, USA

**Correspondence:** Mark Schlutow (mark.schlutow@bgc-jena.mpg.de)

**Abstract.** Understanding how greenhouse gases and pollutants move through the atmosphere is essential for predicting and mitigating their effects. We present a novel atmospheric dispersion and footprint model: the Boundary Layer Dispersion and Footprint Model (BLDFM), which solves the three-dimensional steady-state advection-diffusion equation in Eulerian form using a numerical approach based on the Fourier method, the linear shooting method and the exponential integrator method. In contrast to analytical Gaussian plume or stochastic Lagrangian models, this novel numerical approach proves beneficial as it does not rely on any asymptotic assumptions or estimates. Furthermore, it is fully modular, allowing for the use of a variety of turbulence closure models in its implementation or direct usage of measured or simulated wind profiles. The model is designed to be flexible and can be used for a wide range of applications, including climate impact studies, industrial emissions monitoring and spatial flux attribution. We validate the model using an analytical test case. The numerical results show excellent agreement with the analytical solution. We also compare the model with the well-established Kormann and Meixner (Boundary-Layer Meteorology, 2001) footprint model (FKM) which is based on the analytical Gaussian plume. The results show overall good agreement but some differences in the fetch of the footprints, which are attributed to the neglect of streamwise turbulent mixing – being one of the aforementioned asymptotic assumptions – in the FKM model. Our results demonstrate the potential of the BLDFM model as a useful tool for atmospheric scientists, biogeochemists, ecologists, and engineers.

## 1 Introduction

Accurate quantification of atmospheric transport and mixing of trace gases, particulates, and energy in the planetary boundary layer is crucial for understanding and prediction of pollutant dispersion, greenhouse gas concentrations and fluxes, and surface-atmosphere exchange processes (Stull, 1988; Baldocchi, 2008). Numerical dispersion models are indispensable tools for this task: they integrate meteorological data, chemical processes, and fluid dynamics to simulate how substances are transported and dispersed in the atmosphere. Such models inform a broad range of applications, from air quality management and public health risk assessments to ecosystem evaluations, climate impact studies, industrial emissions monitoring and agricultural planning (Seinfeld and Pandis, 2012). Complementary to dispersion modeling, flux footprint models are essential when inter-



preting measurements taken at fixed observation points, such as eddy covariance towers (Vesala et al., 2008; Aubinet et al.,
2012). The footprint defines the field of view of the measurement point and represents the influence of the surface on the measured fluxes of momentum, heat, moisture, or trace gases, thereby improving our understanding of how surface-atmosphere exchange processes scale from local to regional extents (Leclerc and Foken, 2014). By determining the spatial extent from which measurements arise, flux footprint models provide critical insights for applications in micrometeorology, ecology, and biogeochemistry research (Schmid, 2002; Göckede et al., 2008).

In more recent years, flux footprint models gained attention for application in spatial flux attribution modeling. In this context, the aim is to map the spatial heterogeneity in fluxes of trace gases, moisture, or energy around flux towers. Wang et al. (2006) demonstrated in a case study how $CO_2$ fluxes may be inferred by decomposing flux data from eddy covariance towers using footprint and ecosystem models. Crawford and Christen (2015) as well as Tuovinen et al. (2019) applied a similar technique in combination with environmental controls and surface land cover maps to attribute sources of measured eddy covariance fluxes of $CO_2$ and methane, respectively. Rey-Sanchez et al. (2022) used this approach to detect hot spots of methane. Pirk et al. (2024) exploit a Bayesian Neural Network to spatially disaggregate carbon exchange of degrading permafrost peatlands. The fidelity of these efforts depends on a precise flux footprint model that accurately associates the ground fluxes with the measured fluxes.

Conceptually, both dispersion models and footprint models are based on the transport equation governing the advection and turbulent mixing of a scalar – like temperature, moisture, trace gas or pollutant in the atmosphere. In these models, turbulent mixing is commonly represented through an eddy diffusivity based on K-theory for turbulence closure. Mathematically, the governing equation is a partial differential equation evolving the scalar in space and time, usually referred to as the advection-diffusion equation (Stockie, 2011). On the atmospheric microscale in the planetary boundary layer, the equation can be simplified by assuming a steady state, i.e., by neglecting the time derivative of the scalar. The steady-state assumption is standard in the eddy covariance method ("well-mixed criterion"). It holds due to a scale separation: while advection and turbulent mixing happen predominately on the microscale, the temporal evolution of wind patterns causing this transport occurs, on the other hand, on the mesoscale.

Under a given set of assumptions, the steady-state advection-diffusion equation admits a closed-form analytical solution known as the Gaussian plume. This solution assumes a steady, spatially uniform wind field; steady, isotropic eddy diffusivity; and negligible mixing in the direction of the mean flow. Several dispersion and footprint models are based on similar (semi-)analytical formulations (Pasquill, F., 1972; Schuepp et al., 1990; Lin and Hildemann, 1997; Kormann and Meixner, 2001; Moreira et al., 2005; Krapez and Ky, 2023). However, the asymptotic assumptions necessary to derive analytical closed-form solutions can be quite restrictive, as they are only valid for a limited range of meteorological conditions.

Alternatively, reformulating the advection-diffusion equation such that the frame of reference follows an individual flow parcel yields the Lagrangian specification of the transport problem. The trajectories of the Lagrangian parcels are often modeled stochastically (Hsieh et al., 2000; Kljun et al., 2002, 2015). It should be noted that most stochastic Lagrangian footprint models do not unconditionally satisfy the well-mixed criterion which, as mentioned above, is imperative for the validity of the eddy covariance method (Thomson, 1987).





Instead of parametrizing turbulent mixing by eddy diffusivity, Large Eddy Simulations (LES) resolve the relevant turbulent scales such that advection becomes the dominant transport mechanism for the scalar (Steinfeld et al., 2008; Cai et al., 2010; Schlutow et al., 2024). Despite their capabilities in simulating complex flows over uneven terrain in various turbulent regimes, LES of the planetary boundary layer consist of sophisticated numerical solvers of the Navier-Stokes equations and are computationally expensive.

In this paper, we propose an alternative atmospheric dispersion and footprint model applicable in the planetary boundary layer, which is based on the numerical solution of the three-dimensional advection-diffusion equation in Eulerian form. This numerical solver exploits the Fast Fourier Transform (FFT), allowing for fast and accurate computation. The numerical solution of the transport problem has the following benefits:

1. The well-mixed criterion is unconditionally fulfilled by design.

2. In contrast to analytical approaches, there is no need for asymptotic assumptions or estimates.

3. The accuracy of the solution depends, instead, only on the resolution. Depending on the application, the method's performance may be varied, and the resolution can be chosen to give moderately accurate results at fast computational speed or highly accurate results at moderate speed.

4. Due to its numerical nature, the solver can be implemented in a modular fashion. In particular, the turbulence closure module is separated from the transport solver, allowing for the use of a variety of turbulence models such as first-order, one-and-a-half-order or higher-order closure schemes (Stull, 1988), which may be selected depending on the use case.

5. Alternatively, wind profiles from real-world measurements or LES may be used directly as input.

This paper is structured as follows. Section 2 introduces and derives the novel Boundary Layer Dispersion and Footprint Model (BLDFM). Section 4 revisits Monin-Obukhov Similarity Theory for turbulence closure. In order to assess BLDFM's validity, the numerical solution is tested against a specific analytical solution in Section 5. Section 6 presents a comparison of BLDFM with the widely used and well-established flux footprint model of Kormann and Meixner (2001). Some final remarks are given in Section 7.

## 2  Solving the transport equations: The Boundary Layer Dispersion and Footprint Model (BLDFM)

In this section, we derive the Boundary Layer Dispersion and Footprint Model (BLDFM). As a starting point, we consider the steady-state advection-diffusion equation for some scalar $\Phi$, e.g., temperature, moisture or trace gas concentration, within the planetary boundary layer,

$$\boldsymbol{u}_h(z) \cdot \nabla_h \Phi - K_h(z)\nabla_h^2 \Phi - \frac{\partial}{\partial z}\left(K_z(z)\frac{\partial \Phi}{\partial z}\right) = 0, \tag{1}$$

with flux boundary condition at the surface,

$$-K_z \frac{\partial \Phi}{\partial z} = Q_0(\boldsymbol{x}_h) \text{ at } z = z_0, \tag{2}$$





where $\boldsymbol{x}_h = (x, y)^{\mathsf{T}}$ defines the horizontal coordinates in zonal and meridional direction and $\boldsymbol{u}_h = (u, v)^{\mathsf{T}}$ is its associated

horizontal vector of mean wind. The horizontal nabla operator is given by $\nabla_h = (\partial/\partial x, \partial/\partial y)^{\mathsf{T}}$. Superscript $\mathsf{T}$ denotes the transposed of a vector or matrix. Wind as well as horizontal and vertical eddy diffusivity, $K_h$ and $K_z$, depend solely on the vertical axis $z$. Thus, we model a steady boundary layer flow. The variable $Q_0$ represents the horizontally variable surface kinematic flux of the scalar $\Phi$. It may stand for the sensible heat flux, if $\Phi$ represents temperature, or surface-atmosphere gas exchange flux, if $\Phi$ happens to be a trace gas concentration like $CO_2$ or methane. For the sake of generality, we associate $\Phi$

with arbitrary units (a.u.) symbolizing K, $\mathrm{kg/kg}$, $\mathrm{mol/mol}$ or $\mathrm{kg/m^3}$. Thus, the kinematic flux $Q_0$ has units of $\mathrm{a.u.\,m/s}$.

In the lateral direction, we may assume a periodic (or vanishing) solution. By this assumption, no net flux can be generated over the horizontal boundaries. Since (1) is linear and its coefficients depend only on the z-axis, we may apply the Fourier method in the horizontal direction,

$$\Phi(\boldsymbol{x}_h, z) = \sum_{n=-\infty}^{\infty} \sum_{m=-\infty}^{\infty} \varphi(\boldsymbol{\mu}, z) e^{\mathrm{i}\boldsymbol{\mu} \cdot \boldsymbol{x}_h}, \tag{3}$$

where $\boldsymbol{\mu} = \left(\dfrac{2\pi}{L_x}n, \dfrac{2\pi}{L_y}m\right)^{\mathsf{T}}$, $\tag{4}$

which fulfills the lateral boundary conditions by construction. Inserting the Fourier series (3) into the steady-state advection-diffusion equation (1) transforms the problem into a family of second-order ordinary differential equations with parameters $(n, m)$, which may be written as

$$\frac{\partial}{\partial z}\left(K_z \frac{\partial \varphi}{\partial z}\right) - \sigma^2 \varphi = 0. \tag{5}$$

Equation (5) constitutes an Eigenvalue Problem (EVP) with eigenvalue

$$\sigma^2(\boldsymbol{\mu}, z) = K_h(z)|\boldsymbol{\mu}|^2 + \mathrm{i}\,\boldsymbol{\mu} \cdot \boldsymbol{u}_h(z) \tag{6}$$

and boundary condition

$$-K_z \frac{\partial \varphi}{\partial z} = q_0(\boldsymbol{\mu}) \text{ at } z = z_0. \tag{7}$$

Here, we used the definition of the Fourier coefficients of the surface flux,

$q_0(\boldsymbol{\mu}) = \dfrac{1}{L_x L_y} \displaystyle\int_0^{L_x} \int_0^{L_y} Q_0(\boldsymbol{x}_h) e^{-\mathrm{i}\boldsymbol{\mu} \cdot \boldsymbol{x}_h} \, \mathrm{d}^2 \boldsymbol{x}_h,$ $\tag{8}$

where $[0, L_x] \times [0, L_y]$ depicts the horizontal domain.

We aim to solve the EVP numerically. Hence, we approximate the Fourier series (3) and the integrals in the definition of the Fourier coefficients (8) by finite sums, which allows for usage of Fast Fourier Transform (FFT) when implementing. Consider the solution to (5) on the unbounded domain $z \in [z_0, \infty)$. In order to constrain the solution as $z \to \infty$, we apply the maximum

principle (Evans, 2010, pp 344) that states $\Phi$ must achieve its maximum at the boundary, i.e., at $z = z_0$. The maximum principle



may be implemented by assuming the coefficients of (1) become constant above a certain height $z_M$, which is taken to be the measurement height. By this premise, it is possible to compute an analytical solution $\varphi_M$ for $z > z_M$. The lower boundary condition of the analytical solution for $z > z_M$ eventually becomes the upper boundary condition for the numerical solution for $z_0 \le z \le z_M$. The general solution of (5) with constant coefficients is found to be,

$$\varphi = A + Bz \text{ for } (n,m) = (0,0) \text{ and} \tag{9}$$

$$\varphi = Ce^{-\sigma z} + De^{\sigma z} \text{ for } (n,m) \ne (0,0). \tag{10}$$

Notice that the solution has become degenerate for $(n,m) = (0,0)$. The maximum principle requires that $D = 0$. Hence, the analytical solution for $z > z_M$ is

$$\varphi_{\text{analytic}} = A + Bz \text{ for } (n,m) = (0,0) \text{ and} \tag{11}$$

$$\varphi_{\text{analytic}} = Ce^{-\sigma z} \text{ for } (n,m) \ne (0,0). \tag{12}$$

Taking the aforementioned considerations into account, finding the numerical solution of the EVP with variable coefficients as $z_0 \le z \le z_M$ is transformed into a boundary value problem. For the sake of easier numerical integration, we rewrite (5) as a family of coupled first-order ordinary differential equations by substituting the vertical kinematic flux $q$,

$$\frac{\partial \varphi}{\partial z} = -\frac{q}{K_z(z)},$$
$$\frac{\partial q}{\partial z} = \left(-\mathrm{i}\boldsymbol{\mu} \cdot \boldsymbol{u}_h(z) - K_h(z)|\boldsymbol{\mu}|^2\right)\varphi, \tag{13}$$

with boundary conditions as follows,

$$q = q_0 \text{ at } z = z_0 \text{ for all } (n,m), \tag{14}$$

$$\begin{pmatrix} \varphi \\ q \end{pmatrix} = \begin{pmatrix} \varphi_{\text{analytic}} \\ -K_z \partial \varphi_{\text{analytic}}/\partial z \end{pmatrix} = \begin{pmatrix} A + Bz_M \\ -K_z B \end{pmatrix}$$
$$\text{at } z = z_M \text{ for } (n,m) = (0,0) \text{ and} \tag{15}$$

$$\begin{pmatrix} \varphi \\ q \end{pmatrix} = \begin{pmatrix} \varphi_{\text{analytic}} \\ -K_z \partial \varphi_{\text{analytic}}/\partial z \end{pmatrix} = \begin{pmatrix} Ce^{-\sigma z_M} \\ K_z \sigma Ce^{-\sigma z_M} \end{pmatrix}$$
$$\text{at } z = z_M \text{ for } (n,m) \ne (0,0). \tag{16}$$

Let us consider both cases for $(n,m)$ separately. The boundary conditions depend on free parameters $A, B, C$. Hence, both the ODE and the boundary conditions may be simplified.

For $(n,m) = (0,0)$, $\partial q/\partial z = 0$ and so the flux $q = q_0$ becomes constant, and the problem can be recast as an initial value problem (IVP),

$$\frac{\partial \varphi}{\partial z} = -\frac{q_0}{K_z(z)}, \tag{17}$$

$$\varphi = \varphi_0 \text{ at } z = z_0 \tag{18}$$





where $\varphi_0$ is a new free parameter replacing the role of $A$ and denotes the mean concentration at the surface.

For $(n, m) \neq (0, 0)$, we obtain the following simplification removing the dependency on parameters $B$ and $C$,

$$\frac{\partial \varphi}{\partial z} = -\frac{q}{K_z(z)},$$
$$\frac{\partial q}{\partial z} = \left[ -i\boldsymbol{\mu} \cdot \boldsymbol{u}_h(z) - K_h(z)|\boldsymbol{\mu}|^2 \right] \varphi, \tag{19}$$

$$q = q_0 \text{ at } z = z_0,$$
$$-K_z \sigma \varphi + q = 0 \text{ at } z = z_M. \tag{20}$$

Note that the upper boundary condition is converted into a Robin boundary condition.

This boundary value problem may be solved using the linear shooting method (Lindfield and Penny, 2018). Let $(\varphi_1, q_1)^\mathsf{T}$ be the solution of the initial value problem given by the system of ordinary differential equations (ODEs) (19) and the initial

condition $(1, 0)^\mathsf{T}$ at $z = z_0$. Similarly, let $(\varphi_2, q_2)^\mathsf{T}$ be the solution of the initial value problem with initial condition $(0, q_0)^\mathsf{T}$ at $z = z_0$. The solution to the original boundary value problem, given by (19) and (20) is a linear combination of the two solutions of the initial value problems,

$$\begin{pmatrix} \varphi \\ q \end{pmatrix} = \alpha \cdot \begin{pmatrix} \varphi_1 \\ q_1 \end{pmatrix} + 1 \cdot \begin{pmatrix} \varphi_2 \\ q_2 \end{pmatrix}, \tag{21}$$
$$\alpha = \frac{K_z(z_M)\sigma(z_M)\varphi_2(z_M) - q_2(z_M)}{q_1(z_M) - K_z(z_M)\sigma(z_M)\varphi_1(z_M)}. \tag{22}$$

Notice that the initial values were chosen – without loss of generality – to yield compact expressions for the coefficients of the linear combination.

As the ODE is linear, the IVP may be solved numerically by the exponential integrator method (Hochbruck and Ostermann, 2010). However, methods that are easier to derive and implement likely suffice for most applications. We derived a third-order method from the Taylor expansion of the exponential integrator which proved to be consistent, stable, and robust. It is also

considerably faster than the exponential integrator, rendering it the default method in our implementation. The derivation of the exponential integrator and its third-order approximation can be found in the Appendix A.

## 3  Green's function and footprints

In certain applications, it might be advantageous to solve a slightly altered version of the original problem (1). Eqs. (19) and (20) are written as follows,

$$\frac{\partial G_\Phi}{\partial z} = -\frac{G_Q}{K_z(z)}, \tag{23}$$
$$\frac{\partial G_Q}{\partial z} = K_h(z)\nabla_h^2 G_\Phi - \boldsymbol{u}_h(z) \cdot \nabla_h G_\Phi, \tag{24}$$





with boundary condition

$$-K_z \frac{\partial G_\Phi}{\partial z} = \delta(x)\delta(y) \text{ at } z = z_0. \tag{25}$$

Here, the surface flux source term was substituted with the delta distribution, which essentially represents a unit point source of infinitesimal diameter. The vector consisting of the elements $G_\Phi$ and $G_Q$ is called the Green's function (Finnigan, 2004), and the numerical solution of (23)–(25) follows similarly from Section 2.

Eventually, the solution of the original problem (1) is given by the convolution of the Green's function with the actual surface flux source term,

$$\Phi(\boldsymbol{x}_h, z) = \frac{1}{L_x L_y} \int_0^{L_x} \int_0^{L_y} Q_0(\boldsymbol{\xi}_h) G_\Phi(\boldsymbol{x}_h - \boldsymbol{\xi}_h, z) \, \mathrm{d}^2 \boldsymbol{\xi}_h \tag{26}$$

and similarly

$$Q(\boldsymbol{x}_h, z) = \frac{1}{L_x L_y} \int_0^{L_x} \int_0^{L_y} Q_0(\boldsymbol{\xi}_h) G_Q(\boldsymbol{x}_h - \boldsymbol{\xi}_h, z) \, \mathrm{d}^2 \boldsymbol{\xi}_h. \tag{27}$$

Therefore, in applications where several flux source terms are present (e.g., multitracer approaches) a Green's function needs to be computed only once for a given meteorological condition. The concentration and flux fields are then computed by the convolutions (26) and (27), which become simple sums when discretized. In summary, the approach with Green's function may be computationally much more efficient.

Notice that there is a strong connection between the Green's function and the footprint $F$ (Vesala et al., 2008): the footprint is the reflection of the Green function shifted to the measurement point. In terms of the Green's function, the concentration and flux footprint may be written as,

$$F_\Phi(\boldsymbol{x}_{h,M}, z_M; \boldsymbol{x}_h) = G_\Phi(\boldsymbol{x}_{h,M} - \boldsymbol{x}_h, z_M), \tag{28}$$

$$F_Q(\boldsymbol{x}_{h,M}, z_M; \boldsymbol{x}_h) = G_Q(\boldsymbol{x}_{h,M} - \boldsymbol{x}_h, z_M), \tag{29}$$

respectively.

In conclusion, BLDFM constitutes a dispersion model and a footprint model at the same time.

## 4 Profiles of mean wind and eddy diffusivity

We apply Monin-Obukhov's Similarity Theory (MOST) to obtain the profiles of wind $\boldsymbol{u}_h$ as well as eddy diffusivities $K_h$ and $K_z$ (Monin and Obukhov, 1954; Kormann and Meixner, 2001) taking atmospheric stability into account. Under MOST, the mean horizontal wind speed can be expressed as,

$$u_h(z) = \frac{u_*}{\kappa} \left( \ln\left(\frac{z}{z_0}\right) + \psi_m\left(\frac{z}{L}\right) \right), \tag{30}$$



and the eddy diffusivities, assuming isotropic diffusion, are given by

$$K_h(z) = K_z(z) = \frac{\kappa u_* z}{\phi_c(z/L)}.$$  (31)

The variables $u_*$, $z_0$ and $L$ denote friction velocity, roughness length and Obukhov length, respectively. The latter is a measure of the stability of the boundary layer. They are typical diagnostic variables from eddy covariance measurements. The parameter $\kappa = 0.4$ is the von Kármán constant. Atmospheric stability enters the equations by the universal functions $\psi_m$ and $\phi_c$ which are specified by the Businger–Dyer relationships (Businger et al., 1971; Dyer, 1974) as

$$\psi_m = \begin{cases} 5z/L & \text{for } L > 0, \\ -2\ln\left(\frac{1+\zeta}{2}\right) - \ln\left(\frac{1+\zeta^2}{2}\right) \\ \quad +2\arctan(\zeta) - \frac{\pi}{2} & \text{for } L < 0 \end{cases}$$  (32)

with $\zeta = (1 - 16z/L)^{1/4}$ and

$$\phi_c = \begin{cases} 1 + 5z/L & \text{for } L > 0, \\ (1 - 16z/L)^{-1/2} & \text{for } L < 0. \end{cases}$$  (33)

Since BLDFM is a numerical solver, more options for closure models, like one-and-a-half or second-order closures, can easily be implemented. Furthermore, the choice of isotropic diffusion can straightforwardly be extended by more complex parametrizations.

## 5 Comparison with analytical solution with constant profiles

As shown in (9) and (10) the steady-state advection-diffusion equation admits an analytical solution if its coefficients are constant. This fact provides a practical test case to assess the performance of the numerical method presented in the previous section. Figure 1 shows results from a comparison of the numerical solution with the analytical solution when $u, v, K_h$ and $K_z$ are set constant. The top two plots show the plume of the scalar $\Phi$ and its flux at $z_M = 10\,\text{m}$ caused by a unit point source at the surface marked by a red dot. The bottom two plots show the associated relative differences to the analytical solution as

$$\Delta_{\text{rel}}\Phi = \frac{\Phi_{\text{numerical}} - \Phi_{\text{analytic}}}{\max(\Phi_{\text{analytic}})}.$$  (34)

The wind in the test simulation is set to $\boldsymbol{u}_h = (4.0, 1.0)^\mathsf{T}\,\text{m/s}$ and eddy diffusivity $K_h = K_z = 1.6\,\text{m/s}^2$. These values are chosen to depict typical meteorological conditions in the planetary boundary layer. 512 Fourier modes and 256 vertical grid points are used for the numerical integration with the third-order method as described in Section 2 and Appendix A which might be slightly excessive for usual applications but still is sufficiently fast on a modern desktop computer. Our computer runs an 11th generation Intel®Core™ i7 processor at $2.50\,\text{GHz}$ and has $16\,\text{GiB}$ RAM installed. BLDFM was implemented and compiled with Python 3.13.2. On a single core (unparallelized) the simulation took $9\,\text{s}$.





The results are as expected and a typical plume is generated (cf. Stockie, 2011). The comparison with the analytical solution reveals a remarkably good agreement: the maximum relative difference is less than $1\,‰$. In order to corroborate convergence,

other resolutions and different parameter settings were tested as well. The relative error decreases with higher resolution (not shown here). In conclusion, the numerical integrator of BLDFM demonstrates robust performance in the designed test case scenario, effectively solving the problem with high accuracy and efficiency.

## 6 Comparison with the Kormann and Meixner footprint model

To assess BLDFM's capabilities as a footprint model, we ran two different test cases for various stratification scenarios of the

225 boundary layer and compared the results with the well-established and widely used footprint model of Kormann and Meixner (2001, hereafter referred to as FKM). The FKM model follows a diametrical paradigm compared to BLDFM. In contrast to BLDFM's numerical approach, FKM is based on an analytical closed-form solution of the steady-state advection-diffusion equation subject to two simplifications:

1. The slender plume assumption: turbulent diffusion in streamwise direction is neglected.

2. Vertical profiles of the horizontal wind velocity and eddy diffusivity are modeled in terms of power laws. Profiles from Monin-Obukhov Similarity Theory are then approximated by the power law representation, with reported deviations of up to $15\,\%$.

None of these simplifications are utilized in BLDFM, as the approach is entirely numerical.

Figure 2 presents a comparison of the footprints from BLDFM and FKM under unstable conditions. The wind blows from the

235 North at $6.0\,\mathrm{m/s}$, a roughness length of $z_0 = 0.1\,\mathrm{m}$ is assumed, and the measurement point is at $(x_M, y_M)^\mathsf{T} = (100.0, 0.0)^\mathsf{T}\,\mathrm{m}$ at a height of $z_M = 10.0\,\mathrm{m}$. The Monin-Obukhov length is $L = -20\,\mathrm{m}$ and, hence, the atmospheric conditions are very unstable. At first glance, the footprints of FKM and BLDFM look similar. Both footprints have a comparable fetch, i.e., the distance over which the wind can effectively mix or advect the scalar from its source to the receptor. Upon close inspection it becomes conspicuous, the maximum of the BLDFM footprint is closer to the measurement point in comparison to FKM. This

difference between BLDFM and FKM can be explained by two observations. On the one hand, for very unstable atmospheric conditions, the wind has a strongly sheared profile which is challenging to approximate with a power law in the FKM model. In contrast, BLDFM uses the profiles from MOST directly as input for the numerical integration. On the other hand, FKM neglects streamwise diffusion. Especially in very unstable conditions, eddy diffusivity may be immense, however. Its neglect may cause an underestimation of turbulent transport and therefore an overestimated fetch.

Figure 3 shows a comparison of the footprints from BLDFM and FKM under stable conditions. All features are the same as in the previous case, but the Monin-Obukhov length is chosen to be $L = 20\,\mathrm{m}$, i.e., very stable atmospheric conditions. Both models, BLDFM and FKM, generate footprints with a similar fetch. The main difference between the footprints is the crosswind width, which is recognizably wider for FKM. This difference may be explained by the distinct model choices. In FKM, horizontal eddy diffusivity differs from vertical eddy diffusivity. The latter is determined, similar to BLDFM, by MOST





whereas horizontal eddy diffusivity is modeled in terms of crosswind fluctuation. In contrast, BLDFM assumes isotropic eddy diffusivity. Therefore, BLDFM potentially underestimates horizontal turbulent mixing. In contrast to the test case with very unstable conditions, the difference in fetch is not that pronounced. The minor discrepancy can also be explained by the isotropy of BLDFM's eddy diffusivity, supporting our suspicion that FKM overestimates the fetch due to its neglect of streamwise turbulent mixing.

## 7  Conclusions

This paper presents a novel atmospheric dispersion and footprint model called BLDFM for applications on the microscale in the atmospheric boundary layer for various turbulent regimes dependent on atmospheric stability. The model solves the three-dimensional steady-state advection-diffusion equation in Eulerian form numerically. In contrast to stochastic Lagrangian footprint models, the "well-mixed criterion" is, therefore, unconditionally fulfilled by the steady-state formulation. The Fourier method was utilized to transform the transport problem into a family of second-order ordinary differential equations posing an eigenvalue problem. In combination with flux boundary conditions, the problem was converted into a boundary value problem which can be efficiently solved with the linear shooting method. This numerical formulation allows for usage of the Fast Fourier Transform in combination with the Exponential Integrator Method, resulting in a computationally fast and robust algorithm.

Vertical profiles of mean wind and eddy diffusivity were computed by Monin-Obukhov Similarity Theory, taking the stability of the boundary layer into account. More sophisticated higher-order closure schemes are to be implemented in the future. Notice that vertical wind and diffusivity profiles are represented directly on the grid with user-defined precision which stands in stark contrast to (semi-)analytical footprint models where the profiles need to be fitted to auxiliary functions for which closed-form analytical expressions can be derived. The grid representation also allows to use measured or simulated (e.g. with LES) profiles directly.

BLDFM's performance has been tested against a special analytical solution. Under typical atmospheric conditions and moderately high resolution, the relative difference between numerical and analytical solution is less than 1‰. Comparisons with the Kormann and Meixner footprint model (FKM) for very stable and very unstable conditions showed general agreement. However, discrepancies in the fetch of the footprints were identified. We associated those differences with FKM's neglect of turbulent mixing in the stream direction. This neglect is a necessary assumption in analytical footprint models to obtain closed-form expressions. However, BLDFM takes the full three-dimensional turbulent mixing into account. In a future publication it is planned to validate BLDFM against data from tracer release experiments.

*Code availability.* BLDFM is implemented in Python and freely available on https://github.com/SchlutowSM2Group/BLDFM under GPL-3.0 license. The exact version of BLDFM used to produce the results used in this paper is archived on https://zenodo.org under https://doi.org/10.5281/zenodo.15487244 (Schlutow and Chew, 2025), as are input data and scripts to run the model and produce the plots for all the simulations presented in this paper.



## Appendix A: Derivation of the exponential integrator

In Section 2 we introduced the Exponential Integrator Method to solve the initial value problem arising from the linear shooting method. We discretize the vertical axis by equidistant increments $z^{(n)} = z_0 + n\delta z$ such that for any function $f = \boldsymbol{u}_h, K_h, K_z, \ldots$ we write $f(z^{(n)}) = f^{(n)}$. Then, according to Hochbruck and Ostermann (2010) we may derive the following

iteration scheme in terms of the exponential integrator method,

$$\eta^{(n)} = -K_h^{(n)}|\boldsymbol{\mu}|^2 - \mathrm{i}\,\boldsymbol{\mu} \cdot \boldsymbol{u}_h^{(n)}, \tag{A1}$$

$$\lambda^{(n)} = \sqrt{\eta^{(n)}/K_z^{(n)}}, \tag{A2}$$

$$\varphi^{(n+1)} = \cos\left(\lambda^{(n)}\delta z\right)\varphi^{(n)} - \frac{1}{\lambda^{(n)}K_z^{(n)}}\sin\left(\lambda^{(n)}\delta z\right)q^{(n)}, \tag{A3}$$

$$q^{(n+1)} = \frac{\eta^{(n)}}{\lambda^{(n)}}\sin\left(\lambda^{(n)}\delta z\right)\varphi^{(n)} + \cos\left(\lambda^{(n)}\delta z\right)q^{(n)}. \tag{A4}$$

Note that this solution is exact given that all coefficients are constant in the interval $z^{(n)} < z < z^{(n+1)}$. A less accurate but computationally faster algorithm emerges when taking the third-order Taylor approximation of the exponential integrator, which can be achieved by expanding the cosine and sine as

$$\cos(x) = 1 - \frac{1}{2}x^2 + \mathcal{O}(\|x\|^4), \tag{A5}$$

$$\sin(x) = x - \frac{1}{6}x^3 + \mathcal{O}(\|x\|^4). \tag{A6}$$

*Author contributions.* MS conceptualized the research problem and the initial model formulation. RC contributed to the development and implementation of the initial numerical approach and to discussions shaping the model formulation. MS later refined the model, corrected key issues, and completed its development, validation, and visualization of the results. MS and RC coordinated, implemented, and tested the software code. MG acquired financial support, managed, and supervised the research activity. MS prepared the manuscript with contributions from all co-authors.

*Competing interests.* The authors declare no competing interests.

*Acknowledgements.* RC acknowledges the Deutsche Forschungsgemeinschaft for the funding through the Collaborative Research Center (CRC) 1114 'Scaling cascades in complex systems', project number 235221301, project A02: 'Multiscale data and asymptotic model assimilation for atmospheric flows'. Support was also provided by Schmidt Sciences, LLC, as part of the Climate Modeling Alliance and the Virtual Earth System Research Institute's DataWave project. MS and MG acknowledge the European Research Council (ERC) under

305 the European Union's Horizon 2020 research and innovation program (Grant agreement No 951288, Q-Arctic). We thank Zhan Li for the `fluxfm` code base, which was utilized for the Kormann and Meixner model implementation in this study. We are grateful to David Ho and Saqr Munassar for their helpful comments and valuable discussions.



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



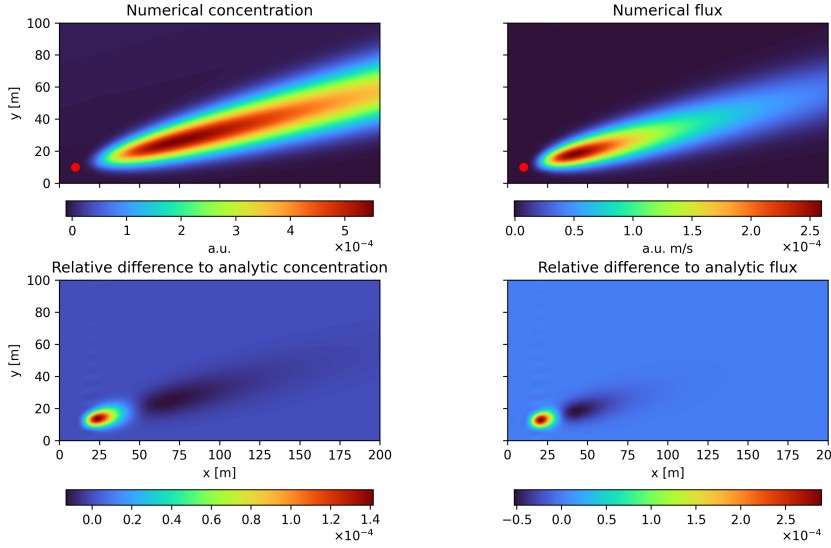

**Figure 1.** Comparison with analytical solution for constant profiles of horizontal wind and eddy diffusivities. Top row: concentration and flux at 10 m computed with the BLDFM numerical scheme arising from a surface unit point source at the red dot. Bottom row: relative differences to the analytic solution.

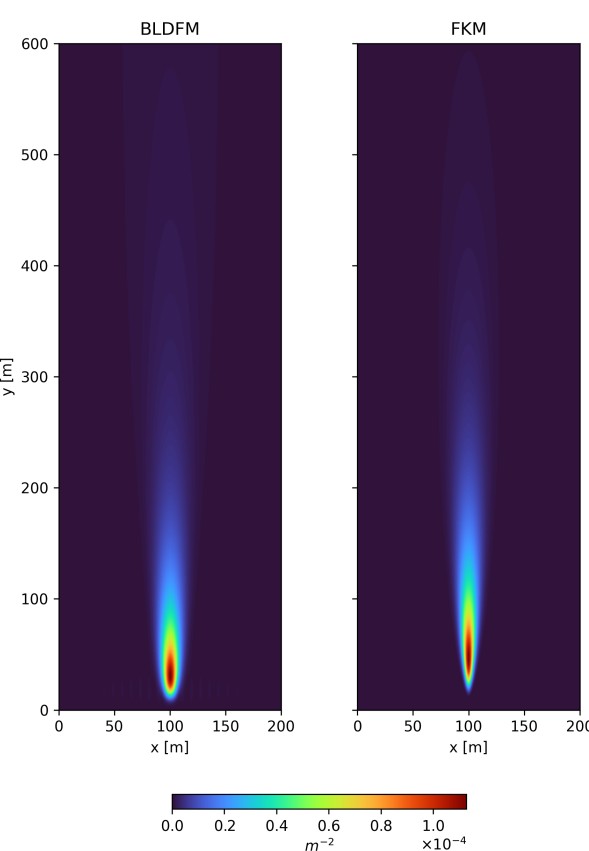

**Figure 2.** Footprints of BLDFM and the Kormann and Meixner footprint model, subject to very unstable conditions of the atmospheric boundary layer (see text for more details).



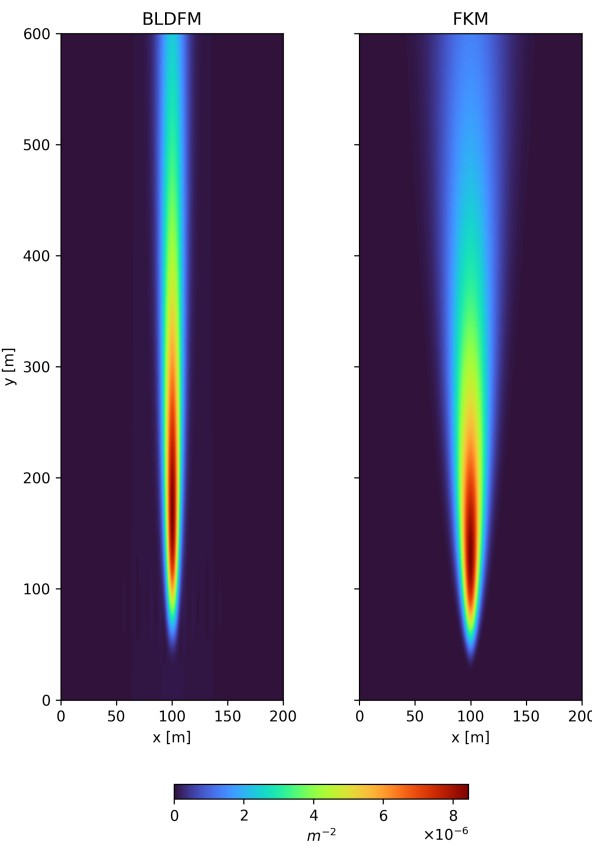

**Figure 3.** Footprints of BLDFM and the Kormann and Meixner footprint model under very stable conditions of the atmospheric boundary layer (see text for more details).