# Peer review of "The Boundary Layer Dispersion and Footprint Model: A fast numerical solver of the Eulerian steady-state advection-diffusion equation"

_EGUsphere, 2025_

## Referee Comment (RC2)

**Review of Manuscript The Boundary Layer Dispersion and Footprint Model: A fast numerical solver of the Eulerian steady-state advection-diffusion equation for GMD**

**Reviewer Report**

**General Comment:**

The manuscript presents a novel atmospheric dispersion and footprint model (BLDFM) that numerically solves the three-dimensional steady-state advection-diffusion equation using a combination of the Fourier method, the linear shooting method, and an exponential integrator. The model is validated against two test cases: one with an available analytical solution, and the other using the established Kormann and Meixner footprint model.

The numerical methodology and implementation details need to be described more thoroughly. I find the work interesting and recommend a major revision before resubmission.

**Major Comments:**

- The manuscript explains the underlying differential equation and the simplifying assumptions. A
  more precise and detailed description of the numerical method and computational results would
  greatly benefit the manuscript:
  - (a) Line 148: The manuscript mentions the linear shooting method to solve the boundary value problems. The authors should explain the method and justify its use in this context.
  - (b) Lines 157–162, Appendix A: The presentation of the numerical method is currently too brief. The manuscript provides limited context as to why this method was chosen and what advantages it offers over alternative numerical approaches. In addition, the authors claim in the main text that the method is consistent, stable, and robust; is there evidence supporting this claim? In the Appendix, is the scheme (A1)–(A4) taken from Hochbruck and Ostermann (2010), and have the authors modified it using the approximations (A5)–(A6)? If so, could they provide readers with a rough estimate of the computational time saved by not evaluating the sine and cosine terms?
  - (c) Section 3: Do the authors implement the Dirac delta distribution mentioned in section 3 in their numerical solver? If so, how is this achieved in practice?
  - (d) Do the authors evaluate the convolutions in Eqs. (26) and (27) in their numerical implementation? If so, how is this done in practice? They might also consider including a short remark about potential parallelization.
  - (e) Section 5: The authors write: "In order to corroborate convergence, other resolutions and different parameter settings were tested as well. The relative error decreases with higher resolution (not shown here)." However, these results should be presented—e.g., in a table or a figure—to allow the reader to assess the convergence behavior quantitatively. Additionally, the BLDFM solver consists of several components, including a Fourier transform, a linear shooting method,

- and an exponential integrator. The statement that "the relative error decreases with higher resolution" is too general. The authors should discuss how each component contributes to the overall numerical error.
- (f) Section 5: The authors should also discuss which parts of the code are amenable to parallelization in order to achieve faster solutions, as this is an important aspect of performance for practical applications.
- (g) Line 248: The authors state: "This difference may be explained by the distinct model choices." It would be helpful to briefly discuss whether numerical errors in BLDFM could contribute to these differences. Ensuring that the observed discrepancies are indeed due to model assumptions rather than numerical artifacts would strengthen the interpretation of the results. The same consideration applies to the unstable case.

**Minor Comments:**

- 1. The authors perform a detailed analytical study of the system. It would be helpful to briefly highlight this contribution in the abstract or introduction, as it provides valuable guidance for the numerical implementation.
- 2. Lines 43, 47: To give the reader a better sense of the scales involved, it would be helpful to provide typical ranges for the atmospheric microscale in the planetary boundary layer and for the mesoscale.
- 3. Lines 45-47: The authors state that advection predominantly occurs on the microscale, while the temporal evolution of wind patterns occurs on the mesoscale. Could the authors clarify whether they refer here specifically to eddy-scale fluctuations rather than mean-flow advection? This would help avoid potential confusion about the scales at which advection acts.
- 4. Line 66: Could the authors comment on whether steep gradients in the scalar field occur in their typical application scenarios, and if so, how the Fourier-based solver deals with them? Since spectral methods may exhibit oscillations near sharp features when resolution is limited, a brief discussion of this aspect would help readers better understand the robustness of the approach.
- 5. Line 96: It would be helpful if the authors could briefly comment on how idealized the assumption of periodic (or vanishing) lateral boundary conditions is. Additionally, do the authors have any thoughts on how this approach could be extended to non-periodic boundary conditions? How would that change the efficiency of the computation of the numerical solution?
- 6. Lines 114ff, Equations (9) (12): For the unbounded domain  $z \in [z_M, \infty]$ , the authors discard the growing exponential term in the solution. This can be interpreted as a physically motivated choice to ensure that the solution remains bounded for large z. It might help to clarify this point and distinguish it from the classical maximum principle, which is formulated for bounded domains. Clarifying this point could help readers better understand the reasoning behind the boundary treatment. In addition, could the authors briefly comment on whether a nonzero coefficient B would be physically meaningful or whether it would necessarily lead to unbounded growth?

- 7. Equation (22): The manuscript provides the formula for  $\alpha$  without explanation. A brief note on how it was computed and how it follows from the linear combination of the two IVP solutions would improve clarity.
- 8. Section 3: It might help the reader if the authors explicitly stated that the Green's function depends on the particular problem.
- 9. Equations (26), (27): The authors might remind the reader that  $Q_0$  is related to the boundary condition or provide a reference to the corresponding equation.
- 10. Section 6: The authors mention that FKM uses simplified assumptions, but it is not clear to the reader which specific equation BLDFM is being compared to. The manuscript would benefit from explicitly stating the governing equations and assumptions of the FKM model to clarify the basis of the comparison.
- 11. Line 270: "BLDFM's performance has been tested against a special analytical solution." Please clarify that this assessment refers to numerical accuracy and not computational runtime or efficiency.

---

## Author Comment (AC1)

**Response to Reviews for the Manuscript "The Boundary Layer Dispersion and Footprint Model: A fast numerical solver of the Eulerian steady-state advection-diffusion equation"**

**Response to Reviewer #1**

We sincerely appreciate the referee's thorough review of our manuscript and the valuable comments and suggestions provided. The feedback has been instrumental in enhancing the quality of our research. We have carefully considered each of the remarks and will accordingly make comprehensive revisions to the manuscript based on the content of the study. Here, we will provide a detailed point-by-point response to all comments, citing the reviewer statements in black font, while our answers are given in blue font.

The validity of the method is verified by selecting a case for which analytical solutions exist. Then an example of an unstable situation is shown where the wind and stability profiles obey Monin-Obukhov similarity. It is compared with a footprint model that uses exponential profiles (FKM). The results are obviously different given that wind profiles are different and that diffusion along the direction of the wind is neglected in FKM. The authors attribute the difference mainly to the missing downwind diffusion, about which I have doubts because diffusion in the direction of the wind is overwhelmed by advection.

We thank the referee for this comment. We agree with the referee that in most situations diffusion is overwhelmed by advection. Nevertheless, we will extend our numerical framework by anisotropic eddy diffusion. This modification allows us to switch off the eddy diffusion component that acts in the direction of the mean flow. With this modification, a direct comparison with the Korman & Meixner model will become possible, which will show that in certain conditions – very unstable atmospheric boundary layer – eddy diffusion can in fact become comparable in its effect to advection.

As a scientific paper, publication would be harder to justify. In that case it would be necessary to simulate a range of cases with more emphasis on the results of the

computational method. Before publication and to show what can be done with the code, I suggest presenting a few more cases, e.g., a stable case and a case with wind turning in the stable boundary layer.

We appreciate the reviewer's comment. To justify a scientific publication, we will add more experiments: in addition to the experiments with the stable and unstable PBL, we will also include the neutrally stratified atmosphere. Two numerical studies will also be incorporated that will prove convergence, challenging the two key components of BLDFM: the Fourier and the Exponential Integrator Methods. Furthermore, we will improve BLDFM by a more precise treatment of the vertical boundary condition and introducing stretched vertical coordinates to further speed up the algorithm.

Wind turning is an important component of dispersion. Vertical diffusion spreads the pollution in the vertical and wind turning amplifies the horizontal spreading of the plume by advection in different directions.

We are thankful for this comment. Wind turning is implicitly accounted for in BLDFM by the turbulent horizontal mixing due to eddy diffusivity. We plan to accommodate this comment by extending the scalar eddy diffusivity acting on the three-dimensional Laplacian to fully three-dimensional vector-like eddy diffusion, which will improve the framework by allowing us to model anisotropic diffusion and simulate, in particular, horizontal diffusion caused by wind veering. A fully dynamic treatment of wind turning goes beyond the capabilities of BLDFM, as the time dependency of the wind is unavailable.

The mention of a "Well-mixed criterion" may be confusing here. In boundary layer meteorology "well mixed" is often used for the convective boundary layer where potential temperature and tracers are fairly constant in the vertical due to strong mixing. Steady state is appropriate near the surface because the mixing time scale is fast compared to evolution of the large scale forcing, often called "quasi-equilibrium". It means that the shape of the profiles are nearly instantaneously in equilibrium with the large scale forcing. The well-mixed criterion is mentioned a few times in the paper. Perhaps it is better to call it the "quasi-equilibrium" assumption throughout.

We appreciate the referee for bringing this issue to our attention. Indeed, the term "well-mixed condition" is misleading and not well explained in the manuscript. The term was coined by Thomson (1987), referring to whether initially well-mixed tracers will remain well-mixed throughout the integration. In a revised version of the manuscript, this will be explained more thoroughly.

To substantiate the conclusion that the difference between BLDFM and FKM is due to differences in diffusion along the wind vector, it would help to run BLDFM without diffusion in the wind direction to see the effect of down wind diffusion.

We are thankful for this comment. As mentioned before, we plan to implement a version of BLDFM without streamwise diffusion for a direct comparison with FKM; the corresponding results will be integrated into this manuscript.

---

## Author Comment (AC2)

**Response to Reviews for the Manuscript "The Boundary Layer Dispersion and Footprint Model: A fast numerical solver of the Eulerian steady-state advection-diffusion equation"**

**Response to Reviewer #2**

We sincerely appreciate the referee's thorough review of our manuscript and the valuable comments and suggestions provided. The feedback has been instrumental in enhancing the quality of our research. We have carefully considered each of the remarks and will make comprehensive revisions to the manuscript based on the content of the study. Next, we will provide a detailed point-by-point response to all comments, citing the reviewer statements in black font, while our answers are given in blue font.

**Major Comments**

The manuscript explains the underlying differential equation and the simplifying assumptions. A more precise and detailed description of the numerical method and computational results would greatly benefit the manuscript:

1. Line 148: The manuscript mentions the linear shooting method to solve the boundary value problems. The authors should explain the method and justify its use in this context.

   We thank the referee for this comment. We will add a paragraph addressing the explanation and justification of the shooting method: "The main idea of the method is to reduce the solution of the BVP to solving two IVPs with arbitrary but linearly independent initial values and reconstruct the BVP solution by a linear combination of the two IVP solutions, allowing for deployment of standard numerical IVP solvers."

2. Lines 157–162, Appendix A: The presentation of the numerical method is currently too brief. The manuscript provides limited context as to why this method was

chosen and what advantages it offers over alternative numerical approaches. In addition, the authors claim in the main text that the method is consistent, stable, and robust; is there evidence supporting this claim? In the Appendix, is the scheme (A1)–(A4) taken from Hochbruck and Ostermann (2010), and have the authors modified it using the approximations (A5)–(A6)? If so, could they provide readers with a rough estimate of the computational time saved by not evaluating the sine and cosine terms?

We appreciate the reviewer's comment. The advantage of the exponential integrator method in comparison to other numerical methods is that it gives exact solutions to linear ODEs with constant coefficients. We will emphasize this fact in a revised version of the manuscript. To provide a more sound investigation of the numerical scheme, we will perform numerical convergence tests and prove exponential error decay, supporting our initial claim of consistency, stability, and robustness. Furthermore, we will improve BLDFM by a more precise treatment of the vertical boundary condition and introducing stretched vertical coordinates to further speed up the algorithm. To evaluate the impact of approximating the sine and cosine terms emerging in the Exponential Integrator Method, we run the same experiment as described in Section 5 but without the Taylor approximation of the sine and cosine terms. Instead of 9 sec, the solver now needed 35 sec. The time difference is substantial.

3. Section 3: Do the authors implement the Dirac delta distribution mentioned in section 3 in their numerical solver? If so, how is this achieved in practice?

We thank the referee for the insightful question. In fact, the Dirac delta function has a simple representation in Fourier space: it is the constant one. This property is used and implemented in BLDFM. It offers the additional advantage that the integral is also precisely one, which is important for mass conservation.

4. Do the authors evaluate the convolutions in Eqs. (26) and (27) in their numerical implementation? If so, how is this done in practice? They might also consider including a short remark about potential parallelization.

We appreciate this question. In the numerical implementation, the Green's function is shifted in Fourier space to the tower location, which amounts to a simple multiplication by a phase. After applying the inverse Fourier transform, the convolution simplifies numerically to a sum over all spatial indices in real space, which is mentioned in the manuscript. BLDFM is indeed parallelized. Please refer to our response to point 6.

5. Section 5: The authors write: "In order to corroborate convergence, other resolutions and different parameter settings were tested as well. The relative error decreases with higher resolution (not shown here)." However, these results should be presented–e.g., in a table or a figure–to allow the reader to assess the convergence behavior quantitatively. Additionally, the BLDFM solver consists of several

components, including a Fourier transform, a linear shooting method, 1 and an exponential integrator. The statement that "the relative error decreases with higher resolution" is too general. The authors should discuss how each component contributes to the overall numerical error.

We are thankful for this comment. In a revised version of the manuscript, we will present additional plots that will show the error convergence for different resolutions. Since BLDFM consists of several numerical methods, mainly the Fourier Method for the horizontal integration and the Exponential Integrator Method for the vertical integration, different test cases will be designed to challenge each component individually. The Fourier method has, theoretically, an exponential error decay; the Taylor approximated exponential integrator should be polynomial of order three. Using the analytical solution as a test case should mostly challenge the Fourier method. Since the vertical profiles are constant, the (not approximated) exponential integrator should be exact. A test case specifically challenging the exponential integrator can be constructed with variable vertical profiles when using a high-resolution simulation as reference. The decay rates will also be measured and reported.

6. Section 5: The authors should also discuss which parts of the code are amenable to parallelization in order to achieve faster solutions, as this is an important aspect of performance for practical applications.

We are grateful for this remark. BLDFM is indeed parallelized. In fact, the vertical integration by the exponential integrator method can be executed independently for each wave number tuple, which makes parallelization straightforward. An explanatory sentence will be added to the manuscript.

7. Line 248: The authors state: "This difference may be explained by the distinct model choices." It would be helpful to briefly discuss whether numerical errors in BLDFM could contribute to these differences. Ensuring that the observed discrepancies are indeed due to model assumptions rather than numerical artifacts would strengthen the interpretation of the results. The same consideration applies to the unstable case.

We appreciate this comment. This section will be rewritten completely in order to make the differences between BLDFM and the Korman & Meixner model much more distinct. We plan to implement a modified version of BLDFM, which also obeys the slender plume assumption, to make the comparison to the FKM model more explicit and direct. To rule out any numerical artifacts, we also plan to perform numerical convergence analyses.

**Minor Comments**

1. The authors perform a detailed analytical study of the system. It would be helpful to briefly highlight this contribution in the abstract or introduction, as it provides valuable guidance for the numerical implementation.

We thank the referee for this comment. However, we do not agree with the reviewer: the analytical solution to the atmospheric transport equation is well known in the literature, which is also cited in the manuscript. Furthermore, the analytical study is mentioned in both the abstract and introduction and does not need further emphasis in our opinion.

2. Lines 43, 47: To give the reader a better sense of the scales involved, it would be helpful to provide typical ranges for the atmospheric microscale in the planetary boundary layer and for the mesoscale.

   We will reformulate the paragraph in question to make it more precise and give typical numbers for the atmospheric scales.

3. Lines 45-47: The authors state that advection predominantly occurs on the microscale, while the temporal evolution of wind patterns occurs on the mesoscale. Could the authors clarify whether they refer here specifically to eddy-scale fluctuations rather than mean-flow advection? This would help avoid potential confusion about the scales at which advection acts.

   Indeed, the term advection is misleading, as it divides into the mean flow and the turbulent component in the Reynolds average. To make the matter clearer, we will emphasize that we consider the Reynolds-averaged equations and add the term 'mean-flow advection' where necessary.

4. Line 66: Could the authors comment on whether steep gradients in the scalar field occur in their typical application scenarios, and if so, how the Fourier-based solver deals with them? Since spectral methods may exhibit oscillations near sharp features when resolution is limited, a brief discussion of this aspect would help readers better understand the robustness of the approach.

   We appreciate this observation. The Fourier method is indeed prone to artificial oscillations at sharp gradients. The extreme case of this Gibbs ringing is when we use the unit point source of infinitesimal diameter to define the footprints. However, in typical use cases, this phenomenon does not pose any issue because the turbulent mixing mathematically represented by the harmonic operator in the form of the Laplacian rapidly smooths out any small-scale oscillations as it is proportional to the wavenumber squared. The benefits of the Fourier method–very fast and conservative–easily outweigh this minor inconvenience.

5. Line 96: It would be helpful if the authors could briefly comment on how idealized the assumption of periodic (or vanishing) lateral boundary conditions is. Additionally, do the authors have any thoughts on how this approach could be extended to non-periodic boundary conditions? How would that change the efficiency of the computation of the numerical solution?

   We thank the reviewer for this observation. Since BLDFM uses the Fourier method, the boundaries are periodic by construction. In order to represent non-periodic domains, a halo of zero flux is used around the domain of interest, effectively

increasing the computation domain. This halo must be sufficiently big such that tracer material that leaves the domain of interest at some edge must not enter at the opposing edge. This treatment of the boundaries makes the algorithm slightly computationally more expensive, but still provides very fast calculations due to the usage of the Fast Fourier Transform. We will add an explanatory paragraph to the manuscript.

6. Lines 114ff, Equations (9) - (12): For the unbounded domain $z \in [z_M, \infty)$, the authors discard the growing exponential term in the solution. This can be interpreted as a physically motivated choice to ensure that the solution remains bounded for large z. It might help to clarify this point and distinguish it from the classical maximum principle, which is formulated for bounded domains. Clarifying this point could help readers better understand the reasoning behind the boundary treatment. In addition, could the authors briefly comment on whether a nonzero coefficient $B$ would be physically meaningful or whether it would necessarily lead to unbounded growth?

   We are grateful for bringing this issue to our attention. The formulation in the manuscript is somewhat imprecise. To be more thorough, actually the weak version of the maximum principle needs to be applied. This will be corrected in the manuscript.

7. Equation (22): The manuscript provides the formula for $\alpha$ without explanation. A brief note on how it was computed and how it follows from the linear combination of the two IVP solutions would improve clarity.

   We appreciate the comment. We will add an explanatory sentence: "Notice that the initial values were chosen–without loss of generality–to yield compact expressions for the coefficients of the linear combination, which are computed by accounting for the boundary values at $z_T$."

8. Section 3: It might help the reader if the authors explicitly stated that the Green's function depends on the particular problem.

   We agree with the referee and will improve the manuscript accordingly by calling it the Green's function of atmospheric dispersion.

9. Equations (26), (27): The authors might remind the reader that $Q_0$ is related to the boundary condition or provide a reference to the corresponding equation.

   The reference to equation (2) will be added to the manuscript together with an explicit mention of the variable $Q_0$.

10. Section 6: The authors mention that FKM uses simplified assumptions, but it is not clear to the reader which specific equation BLDFM is being compared to. The manuscript would benefit from explicitly stating the governing equations and assumptions of the FKM model to clarify the basis of the comparison.

We thank the reviewer for bringing this lack of explanation to our attention. This paragraph in question will be rewritten substantially. As mentioned earlier, we plan to implement a modified version of BLDFM that also obeys the slender plume assumption to make the comparison to the FKM model more explicit and direct.

11. Line 270: "BLDFM's performance has been tested against a special analytical solution." Please clarify that this assessment refers to numerical accuracy and not computational runtime or efficiency.

    We will rephrase this sentence, making the results of the assessment more clear.